# An Inverse Analysis for Establishing the Temperature-Dependent Thermal Conductivity of a Melt-Cast Explosive across the Whole Solidification Process

**DOI:** 10.3390/ma15062077

**Published:** 2022-03-11

**Authors:** Lei Ni, Xiangrong Zhang, Lin Zhou, Xiufen Yang, Bo Yan

**Affiliations:** 1State Key Laboratory of Explosion Science and Technology, Beijing Institute of Technology, Beijing 100081, China; 3120185174@bit.edu.cn (L.N.); zhoulin@bit.edu.cn (L.Z.); y18801328015@gmail.com (X.Y.); 2Research Institute of Gansu Yinguang Chemical Industry Group, Baiyin 730900, China; yanb6825@gmail.com

**Keywords:** melt-cast explosives, temperature-dependent thermal conductivity, inverse heat-transfer problem, Gauss–Newton algorithm

## Abstract

Thermal conductivity is one of the most important thermophysical properties of a melt-cast explosive. However, the temperature-dependent thermal conductivity of such explosives cannot be easily measured across the whole solidification process (including the liquid, semi-solid, and solid states). This study used an inverse analysis method to estimate the temperature-dependent thermal conductivity of a 2,4-dinitroanisole/cyclotetramethylenetetranitramine (DNAN/HMX) melt-cast explosive in a continuous way. The method that was used is described here in detail, and it is verified by comparing the estimated thermal conductivity with a prespecified value using simulated measurement temperatures, thereby demonstrating its effectiveness. Combining this method with experimentally measured temperatures, the temperature-dependent thermal conductivity of the DNAN/HMX melt-cast explosive was obtained. Some measured thermal conductivity values for this explosive in the solid state were used for further validation.

## 1. Introduction

In the manufacturing of melt-cast explosives, a high-temperature suspension of molten explosives begins to cool down until it finally solidifies at room temperature. During this solidification process, there is a significantly inhomogeneous temperature distribution inside the explosive charge and there are notable temperature gradients [1]. This could result in severe thermal stress and cause cracking or damage if the mechanical strength of the explosive is not high enough to withstand the resulting forces [2,3]. To reduce or eliminate such thermal cracking and damage, the thermophysical and/or mechanical properties of the explosive charge need to be improved.

As a key thermophysical property, thermal conductivity has been widely investigated to enhance the thermal-environment adaptability of brittle materials [4,5,6,7]. Before investigating the effects of thermal conductivity on the thermal safety of these brittle materials, their temperature-dependent thermal conductivity should be experimentally measured over as wide a temperature range as possible. Recently, a steady-state hot-wire method has been proposed to measure the thermal conductivity of liquids [8], and an unconventional laser flash technique method has been used to measure solids such as CFRP [9]. However, the temperature-dependent thermal conductivities of materials in their liquid and solid states are generally measured separately using commercially available instruments [10], and the thermal conductivity in the semi-solid state has to be interpolated in some way. Although Sandia National Laboratories recently developed a new method combining finite-element analysis with cookoff data to determine the temperature-dependent thermal conductivity of melt-cast explosives [11], their method involves the complicated solution of Navier–Stokes equations, and it thus cannot be easily followed. Therefore, there is still a need for a simple method for obtaining the temperature-dependent thermal conductivity of melt-cast explosives continuously across the whole solidification process.

Inverse heat transfer problems (IHTPs) have been widely accepted and used as alternative or essential approaches in many engineering applications [12,13,14]. IHTPs originally emerged in the late 1950s when it was found that the aerodynamic heating of space vehicles is so high during re-entry to the atmosphere that the surface temperature of the thermal shield cannot be measured directly using temperature sensors. Since then, IHTPs have been used to estimate all kinds of unknown quantities [14,15,16,17,18,19,20,21], including, among others, initial/boundary conditions, source terms, geometry, heat flux, and thermophysical properties.

The application of IHTPs for establishing thermophysical properties usually involves the estimation of temperature-dependent thermal conductivity and/or specific heat capacity [22,23]. Generally, both the thermal conductivity and specific heat capacity can be identified by a parameter-estimation or function-estimation method [14], and both of these methods can be either deterministic or stochastic. However, both methods involve minimization of the sum of the squares of the difference between the experimental and computational (recovered from an inverse method) temperatures [14]. It should be noted that although such inverse analysis methods have achieved much success in the estimation of the temperature-dependent thermal conductivity of inert materials excluding the solidification process, few studies to date have investigated energetic materials such as melt-cast explosives across the whole solidification process (including the liquid, semi-solid, and solid states).

This study focused on the estimation of the temperature-dependent thermal conductivity of a 2,4-dinitroanisole/cyclotetramethylenetetranitramine (DNAN/HMX) melt-cast explosive during the whole solidification process using an inverse analysis method. The melt-cast explosive examined consists of 70 wt% HMX, 29.5 wt% DNAN, and 0.5 wt% *N*-methyl-4-nitroaniline (MNA). The small amount of MNA was used as a processing agent to lower both the melting point of DNAN and the viscosity of the molten explosive suspension [24]. The explosive performance of this formulation was designed to be comparable to that of Octol (65 wt% HMX and 35 wt% trinitrotoluene (TNT)) [25], but the explosive insensitivity of this formulation is significantly better than that of the Octol formulation.

With the help of other known temperature-dependent thermophysical properties (density and specific heat capacity), the Gauss–Newton algorithm was used in the inverse analysis method to minimize the sum of the squares of the difference between the experimental and computational temperatures. In the remainder of this paper, the experiments and the inverse analysis method will be described in detail. After the inverse analysis method was verified, it was used to estimate the thermal conductivity of the DNAN/HMX melt-cast explosive. Partial validation of the inverse analysis method and factors affecting the estimation are also discussed.

## 2. Experiments and Inverse Analysis Method

### 2.1. Experimental Design

The aforementioned inverse analysis method requires the experimental measurement of temperature profiles. The temperature-dependent density and specific heat capacity of the DNAN/HMX melt-cast explosive also need to be measured for this analysis.

Figure 1 shows a schematic of the experimental setup for measuring the temperature profiles of different parts of the sample. This consists primarily of a casting-mold system and a thermocouple temperature-measurement system. The casting-mold system comprises a cylindrical mold with an inner diameter of 50 mm, an outer diameter of 80 mm, a height of 250 mm, and a sealed cover. The mold is made from 45${}^{\#}$ steel (0.42–0.50 wt% C, 0.50–0.80 wt% Mn, ≤0.035 wt% P, ≤0.035 wt% S). Four K-type thermocouples are embedded at different circumferential and radial positions in the explosive charge; these positions are located radially at 5 mm (A1 and A3) and 21 mm (A2 and A4) from the center of the explosive cylinder. Adjacent thermocouples differ in their positions by 90${}^{\circ }$ in the circumferential direction. The temperatures recorded at the two positions (5 and 21 mm from the center of the cylinder) are approximately one-dimensional and differ only in their radial direction.

The experimental procedures involved the following steps:The molten DNAN/HMX explosive was prepared with an initial temperature of 100.1 ${}^{\circ }C$.The molten explosive was poured into the mold and the sealing cover was placed immediately on the top of the mold and held securely in place using tightening bolts.The molten explosive began to cool, and the four thermocouples recorded the temperature time history until the temperature had decreased to reach that of the ambient environment.

The density of the solid-state explosive charge at normal temperature (21.1 ${}^{\circ }C$) was measured by hydrostatic weighing with an electronic balance. The density of the molten explosive at high temperature (100.1 ${}^{\circ }C$) was measured by the following procedure:The molten explosive was poured into a steel mold with an inner diameter of 25 mm and a height of 100 mm, ensuring that the mold was completely filled so the volume (*V*) of the molten explosive was roughly equal to that of the mold.The mass (*M*) of the molten explosive was measured with a balance after it had solidified and cooled down to normal temperature.The density (ρ) of the molten explosive was evaluated according to ρ=M/V.

The temperature-dependent specific heat capacity of the DNAN/HMX melt-cast explosive could then be determined by differential scanning calorimetry (DSC). In this study, the temperature-dependent thermal conductivity of this explosive during the whole solidification process was estimated by an inverse analysis method. Besides, the thermal conductivity values in the solid state were also measured using a C-Therm (TCi) thermal conductivity analyzer, and the results were used to partially validate the present method.

### 2.2. Inverse Analysis of Thermal Conductivity

In contrast to the function-estimation method, if some information is available on the functional form of the unknown quantities, then the parameter-estimation method only requires the estimation of a few unknown parameters. Using the parameter-estimation method, we assume that the temperature-dependent thermal conductivity k(T) of the DNAN/HMX melt-cast explosive can be represented as a polynomial in the form [26,27]
(1)$$k\left(T\right)=\sum_{j=1}^{N}K_{j}T^{j-1},$$
where *k* is the thermal conductivity, *T* is the temperature, $K_{j}$(j=1,⋯,N) are unknown polynomial parameters, and *N* is the number of unknown parameters. The inverse analysis of the unknown function k(T) is then reduced to the problem of estimating a limited number of parameters $K_{j}$.

The solution of this IHTP for the estimation of the *N* unknown parameters $K_{j}$ is based on the minimization of the least-squares norm (in matrix form)
(2)$$S\left(K\right)={\left[Y-T\left(K\right)\right]}^{T}\left[Y-T\left(K\right)\right],$$
where the superscript T denotes the matrix transpose, Y is the vector of experimental temperatures, K is the vector of unknown parameters $K={\left[K_{1}\;K_{2}\;\cdots \;K_{N}\right]}^{T}$, and T(K) is the vector of computational temperatures at the experimental measurement location.

The minimization of Equation (Equation 2) is implemented by a Gauss–Newton algorithm, which mainly involves the solution of a direct problem and a sensitivity problem.

#### 2.2.1. Direct Problem

In cylindrical coordinates, the one-dimensional transient heat-conduction equations with a phase change can be written as [28]
(3)$$\begin{array}{cccc} \rho \left(T\right)\frac{\partial H\left(r,t\right)}{\partial t} & =\frac{1}{r}\frac{\partial }{\partial r}\left[rk\left(T\right)\frac{\partial T\left(r,t\right)}{\partial r}\right] & \; & \mathrm{for}\;0<r<r_{0},\;t>0,\\ \frac{\partial T\left(r,t\right)}{\partial r} & =0 & \; & \mathrm{for}\;r=0,\;t>0,\\ T\left(r,t\right) & =f\left(t\right) & \; & \mathrm{for}\;r=r_{0},\;t>0,\\ T\left(r,0\right) & =g\left(r\right) & \; & \mathrm{for}\;0\leq r\leq r_{0},\;t=0, \end{array}$$
where *r* is the distance from the origin, *t* is time, $r_{0}$ is the outer radius of the one-dimensional cylindrical geometry, and *H* is the enthalpy of the explosive. We assume linear release of latent heat over the temperature range $T_{s}\leq T\leq T_{\mathrm{liq}}$, where $T_{s}$ and $T_{\mathrm{liq}}$ are the solidus and liquidus temperatures, respectively. The variation of *H* with temperature can be expressed as
(4)$$H=\left\{\begin{array}{ccc} \int_{T_{0}}^{T}c\left(T\right)\;dT & \; & \mathrm{if}\;T<T_{s},\\ \int_{T_{0}}^{T}c\left(T\right)\;dT+\frac{T-T_{s}}{T_{\mathrm{liq}}-T_{s}}l & \; & \mathrm{if}\;T_{s}\leq T\leq T_{\mathrm{liq}},\\ \int_{T_{0}}^{T}c\left(T\right)\;dT+l & \; & \mathrm{if}\;T>T_{\mathrm{liq}}, \end{array}\right.$$
where *c* is the specific heat capacity of the explosive charge and *l* is the latent heat of solidification associated with the phase change of the molten explosive.

Since the thermophysical properties (density, specific heat capacity, and thermal conductivity) are temperature dependent and/or a phase change exists, the direct problem (Equation (Equation 3)) is nonlinear without an analytical solution, and it can be numerically solved iteratively. Given the thermophysical properties and the initial and boundary conditions, the direct problem can be solved using the standard enthalpy method [29,30,31].

#### 2.2.2. Sensitivity Problem

The core of the sensitivity problem is to determine the elements in the sensitivity matrix J (also called the Jacobian matrix). The elements of this matrix (called the sensitivity coefficients) $J_{ij}$ represent the sensitivity of the estimated temperature $T_{i}$ with respect to changes in the unknown parameter $K_{j}$; that is,
(5)$$J_{ij}=\frac{\partial T_{i}}{\partial K_{j}},$$
where the subscript *i* indicates time $t_{i}$(i=1,2,⋯,I) and j=1,2,⋯,N.

Generally, the sensitivity coefficients can be obtained in one of three ways: using the analytical, finite-difference, or boundary-value methods [14]. Since the direct problem is nonlinear, the analytical method is not available. Additionally, the finite-difference method can be very time consuming. Therefore, the boundary-value method was adopted for determining the sensitivity coefficients.

The governing equations of the sensitivity problem can be obtained by differentiating the original direct problem with respect to the unknown parameters. After Equation (Equation 1) is substituted into Equation (Equation 3), the derivative $\partial /\partial K_{j}$ is taken on both sides of the equations. Then, we have
(6)$$\begin{array}{cc} \rho \left(T\right)c\left(T\right)\frac{\partial J_{j}\left(r,t\right)}{\partial t} & =k\left(T\right)\frac{\partial^{2}J_{j}\left(r,t\right)}{\partial r^{2}}+{\left(\frac{\partial T}{\partial r}\right)}^{2}\frac{\partial^{2}k\left(T\right)}{\partial T\partial K_{j}}\\ \; & \;+\left[\frac{\partial k\left(T\right)}{\partial T}\frac{\partial T}{\partial r}+\frac{k\left(T\right)}{r}\right]\frac{\partial J_{j}\left(r,t\right)}{\partial r}\\ \; & \;+\left[\frac{\partial^{2}T}{\partial r^{2}}+\frac{1}{r}\frac{\partial T}{\partial r}\right]\frac{\partial k\left(T\right)}{\partial K_{j}}\;\mathrm{for}\;0<r<r_{0},\;t>0,\\ \frac{\partial J_{j}\left(r,t\right)}{\partial r} & =0\;\mathrm{for}\;r=0,\;t>0,\\ J_{j}\left(r,t\right) & =0\;\mathrm{for}\;r=r_{0},\;t>0,\\ J_{j}\left(r,t\right) & =0\;\mathrm{for}\;0\leq r\leq r_{0},\;t=0, \end{array}$$
where $J_{j}=\partial T/\partial K_{j}$(j=1,2,⋯,N). Combining $J_{j}$ with $J_{ij}$ (as defined in Equation (Equation 5)), it can be seen that $J_{ij}=J_{j}\left(r_{\mathrm{meas}},t_{i}\right)$, where transient temperature measurements are taken at location $r=r_{\mathrm{meas}}$ and at time $t=t_{i}$.

If k(T), ρ(T), c(T), and *T* are known, Equation (Equation 6) is linear and can be directly solved by the finite-difference method. The sensitivity problem needs to be solved *N* times to compute the sensitivity coefficients with respect to each parameter $K_{j}$(j=1,2,⋯,N).

#### 2.2.3. Gauss–Newton Algorithm for Minimization

The Gauss–Newton algorithm was used to minimize the least-squares norm (S(K) in Equation (Equation 2)). Compared to Newton’s algorithm, the Gauss–Newton algorithm has the advantage that it does not require the computation of second derivatives [32]. The necessary conditions for the minimization of S(K) require that the gradient of S(K) with respect to the vector of parameters K must be zero, i.e.,
(7)$$-2J^{T}\left(K\right)\left[Y-T\left(K\right)\right]=0,$$
where the superscript T denotes the matrix transpose, and J is the Jacobian matrix (Equation (Equation 5)), which can be constructed by numerically solving the sensitivity problem (Equation (Equation 6)).

The vector of temperatures (T(K) in Equation (Equation 7)) can be further linearized by a Taylor-series expansion,
(8)$$T\left(K\right)=T\left(K^{n}\right)+J^{n}\left(K-K^{n}\right),$$
where $T(K^{n})$ and $J^{n}$ are the estimated temperatures and the sensitivity matrix evaluated at iteration *n*, respectively. Substituting Equation (Equation 8) into Equation (Equation 7) and rearranging the resulting expression, the iterative procedure to obtain the vector of unknown parameters K can be given by
(9)$$K^{n+1}=K^{n}+{\left[{\left(J^{n}\right)}^{T}J^{n}\right]}^{-1}{\left(J^{n}\right)}^{T}\left[Y-T\left(K^{n}\right)\right].$$

#### 2.2.4. Stopping Criterion

The iterative procedure (Equation (Equation 9)) is not stopped until the criterion
(10)$$S\left(K^{n+1}\right)<\varepsilon$$
is satisfied, where ε can be chosen as a sufficiently small number for errorless measurements. However, errorless measurements are generally impossible. When measurement error is involved, the discrepancy principle is used to stop the iterative procedure [14]. In this case, it will be stopped when the residuals between the measured and estimated temperatures are of the same order of magnitude as the measurement errors. The temperature residuals might be approximated by
(11)$$|Y\left(t_{i}\right)-T\left(r_{\mathrm{meas}},t_{i}\right)|≅\sigma ,$$
where σ is the standard deviation of the measurements, and this is assumed to be a constant. Substituting Equation (Equation 11) into Equation (Equation 2), the value of the stopping criterion ε can be given by
(12)$$\varepsilon =I\sigma^{2}.$$

#### 2.2.5. Computational Procedure

The iterative procedure for estimation of the unknown parameters K is shown in Figure 2. Data including the temperature-dependent density (ρ(T)) and specific heat capacity (c(T)), the solidus and liquidus temperatures, and the latent heat are inputted in advance. The present algorithm was implemented using MATLAB.

## 3. Results and Discussion

### 3.1. Experimental Temperatures and Thermophysical Properties

Figure 3 shows the experimental temperature profiles at the positions r=5 mm ($A_{1}$ and $A_{3}$ in Figure 1) and r=21 mm ($A_{2}$ and $A_{4}$ in Figure 1). The two temperature profiles at the same distance (r=5 or 21 mm) almost overlap, indicating that the position errors for the thermocouple sensors were negligibly small. Additionally, since the inner diameter (50 mm) of the mold was small, the temperature profiles (r=21 mm) near to the inner surface of the mold dropped rapidly, and their initial values were only about 81 ${}^{\circ }C$, while the initial temperature of the molten explosive was 100.1 ${}^{\circ }C$. Compared to the temperature profiles at r=21 mm, the temperature profiles at r=5 mm (near to the center of the mold) dropped gradually. This is due to the latent heat of the liquid–solid phase transition, and the latent heat is released slowly because the measuring points ($A_{1}$ and $A_{3}$ in Figure 1) are far away from the wall of the mold.

The temperature-dependent density and specific heat capacity of the DNAN/HMX melt-cast explosive can be described by Equations (Equation 13) and (Equation 14), respectively:(13)$$\rho \left(\mathrm{kg}/m^{3}\right)=1765-1.25T\;\left(20{\;}^{\circ }C\leq T\leq 100{\;}^{\circ }C\right),$$
(14)$$c\left[J/\left(\mathrm{kg}\cdot K\right)\right]=1240+2T\;\left(20{\;}^{\circ }C\leq T\leq 100{\;}^{\circ }C\right).$$

The densities of the explosive at 100.1 ${}^{\circ }C$ (liquid state) and 21.1 ${}^{\circ }C$ (solid state) were respectively measured as 1640 and 1740 kg/m${}^{3}$, clearly complying with the rule of a common material that expands when heated and contracts when cooled. The density values of the explosive at temperatures greater than 21.1 ${}^{\circ }C$ and less than 100.1 ${}^{\circ }C$ were obtained by linear interpolation. Based on the DSC curve of the heat flow, the dependence of the specific heat capacity of the explosive on the temperature can also be approximated by a linear function. However, it should be noted that the specific heat capacity of the explosive at temperatures above the solidus line and below the liquidus line were not obtained directly from the DSC curve due to the lack of physical meaning, rather, these were also interpolated linearly.

As shown in Figure 4, the measured thermal conductivity of the DNAN/HMX melt-cast explosive in the solid state decreases with increasing temperature. However, this temperature effect is not prominent since the relative difference between the maximum (0.398 W/(m·K)) and the minimum (0.374 W/(m·K)) is only about 6%.

### 3.2. Verification of Inverse Analysis Method

The inverse analysis method was verified using two types of prespecified functions for temperature-dependent thermal conductivity. In type one, the thermal conductivity varies linearly with temperature; in type two, the thermal conductivity varies with temperature in a quadratic polynomial form. Furthermore, for either type of function, a simulated measurement temperature was used. This temperature was generated by combining the solution of the direct problem (to obtain an exact temperature) with the addition of random noise to the exact temperature [14]. The direct problem was solved with the prespecified thermal conductivity function, while the random noise was ωσ, where ω is a random variable with a value of between −2.576 and 2.576 for a 99% confidence bound and σ is the standard deviation of the measurement (assumed to be the same for all measurements).

When solving the direct problem (Equation (Equation 3)), the geometry, initial/boundary conditions, and thermophysical properties are the same as those in the practical solidification/cooling of the DNAN/HMX melt-cast explosive (with respect to Figure 1), except for the prespecified temperature-dependent thermal conductivity. Although the radius of the explosive charge was 25 mm in the approximately one-dimensional cylindrical geometry, the actual radius used in the computation was 21 mm ($A_{2}$ and $A_{4}$ in Figure 1), and the measured temperature profile was treated as a boundary condition. The total computational time t=3000 s. The time step dt=2 s and mesh size dr=0.21 mm; this allowed a good balance between computational efficiency and accuracy. Initial temperatures between r=5 mm and r=21 mm were interpolated linearly. The latent heat *l* was 29.5 kJ/kg, and the solidus ($T_{s}$) and liquidus ($T_{\mathrm{liq}}$) temperatures were 62 ${}^{\circ }C$ and 88 ${}^{\circ }C$, respectively. The temperature-dependent density and specific heat capacity are given by Equations (Equation 13) and (Equation 14), respectively. As mentioned earlier, the two types of prespecified thermal conductivity are given by Equations (Equation 15) and (Equation 16):(15)$$k\left[W/\left(m\cdot K\right)\right]=0.26-5\times 10^{-4}T\;\left(20{\;}^{\circ }C\leq T\leq 100{\;}^{\circ }C\right),$$
(16)$$k\left[W/\left(m\cdot K\right)\right]=0.2625-6.5\times 10^{-4}T+2.64\times 10^{-6}T^{2}\;\left(20{\;}^{\circ }C\leq T\leq 100{\;}^{\circ }C\right).$$

Figure 5 and Figure 6 show the effect of random noise on the recovered temperatures and on the estimated temperature-dependent thermal conductivity. It can be seen that the random noise has a negligible effect on the recovered temperature curves (Figure 5a and Figure 6a), even if the standard deviation ($\sigma =0.5{\;}^{\circ }C$) is much larger than normally expected in precise temperature measurements. However, the random noise does have an appreciable effect on the estimated thermal conductivity (Figure 5a and Figure 6b). Figure 7 further examines this effect on the basis of relative average error, which is defined as [27]:
(17)$$E_{\mathrm{ra}}=\frac{1}{N_{t}}\sum_{i=1}^{N_{t}}\left|\frac{k\left(T_{i}\right)-\hat{k}\left(T_{i}\right)}{\hat{k}\left(T_{i}\right)}\right|\times 100\%,$$
where k(T) and $\hat{k}\left(T\right)$ are the estimated and prespecified values of thermal conductivity, respectively, and $N_{t}$ is the total number of temperature interpolation points to calculate the relative average error. As shown in Figure 7, the maximum relative average error is less than 0.5%, demonstrating that the present inverse analysis method is applicable to the estimation of temperature-dependent thermal conductivity with the help of experimental temperatures.

### 3.3. Estimation of Temperature-Dependent Thermal Conductivity

After the verification using the simulated measurement temperatures with a prespecified thermal conductivity, the present inverse analysis method was used to estimate the temperature-dependent thermal conductivity of the DNAN/HMX melt-cast explosive with the help of the experimentally measured temperatures. A third-order polynomial (*N* is 4 in Equation (Equation 1)) was used to approximate the temperature-dependent thermal conductivity. The total computational time was taken as 1500 s since a larger computational time would make temperature profiles $A_{1}$ (Figure 3, used in the inverse analysis) and $A_{2}$ (Figure 3, serving as a boundary condition) tend to coincide, resulting in considerable inaccuracy when estimating the thermal conductivity [14].

As shown in Figure 8a, the recovered and experimental temperatures are in very good agreement. Below the solidus line (62 ${}^{\circ }C$), the temperature has a nonsignificant effect on the estimated thermal conductivity, and good agreement also exists between the estimated and measured values (Figure 4) of the thermal conductivity (Figure 8b). However, above the solidus line, the estimated thermal conductivity is affected greatly by the temperature, and it increases with increasing temperature. Furthermore, if the temperature profiles $A_{3}$ (r=5 mm, the same as $A_{1}$) and $A_{4}$ (r=21 mm, the same as $A_{2}$), instead of $A_{1}$ and $A_{2}$, are used to estimate the thermal conductivity, basically the same results can be obtained (Figure 9). The estimated temperature-dependent thermal conductivity for the DNAN/HMX melt-cast explosive during the whole solidification process can be expressed as
(18)$$\begin{array}{c} k\left(T\right)=-0.6651+9.4186\times 10^{-2}T-2.3696\times 10^{-3}T^{2}+1.8228\times 10^{-5}T^{3},\\ \left(20{\;}^{\circ }C\leq T\leq 100{\;}^{\circ }C\right). \end{array}$$

Based on the previous analysis, it seems that the temperature-dependent thermal conductivity of the DNAN/HMX melt-cast explosive can be uniquely expressed by Equation (Equation 18). However, it should be noted that the lowest experimental temperature value used in the estimation of thermal conductivity was about 40 ${}^{\circ }C$ (Figure 3 and Figure 8a), which implies that the agreement between the estimated and measured thermal conductivities when the temperature is below 40 ${}^{\circ }C$ may only be coincidental. To further illustrate this, second-order (*N* is 3 in Equation (Equation 1)) and fourth-order (*N* is 5 in Equation (Equation 1)) polynomials were additionally used to approximate the temperature-dependent thermal conductivity.

Figure 10 examines the effect of the polynomial order on the recovered temperatures and on the estimated thermal conductivity. Although the recovered temperatures are in good agreement (Figure 10a), the estimated thermal conductivity varies greatly for different polynomial orders (second, third, and fourth) when the temperatures are below 40 ${}^{\circ }C$ or above 90 ${}^{\circ }C$ (Figure 10b). Clearly, the second-order polynomial is not suitable for approximating the temperature-dependent thermal conductivity; the order is too low. However, even for higher-order polynomials (third- and fourth-order polynomials, as shown in Figure 10b), there is still uncertainty in the estimated thermal conductivity. When the temperatures are below 40 ${}^{\circ }C$, this uncertainty can be attributed to the aforementioned reason (the experimental temperature values used in the estimation of thermal conductivity were all above 40 ${}^{\circ }C$), while the uncertainty above 90 ${}^{\circ }C$ can be explained by the sensitivity coefficient.

Generally, when using an inverse analysis method, the sensitivity coefficient should be large enough for an accurate estimation of a certain physical quantity [14]. However, taking the third-order polynomial as an example, the sensitivity coefficient is zero for every unknown parameter at the beginning of the iterative calculation (Figure 11). Therefore, there must be some uncertainty in the estimated thermal conductivity at the initial stage of the high-temperature liquid state of the molten explosive.

However, when the temperature is above the solidus line, all the estimated thermal conductivities increase with increasing temperature (Figure 10b), no matter the order of the polynomial. This is because the cooling rate is largely affected by the thermal diffusion coefficient α (which is equal to k/ρc), and the greater the thermal diffusion coefficient, the faster the cooling. The temperature profile drops rapidly (large cooling rate) at the beginning of cooling and solidification (Figure 3), while the product of the density and the specific heat capacity is almost constant (Equations (Equation 13) and (Equation 14)); therefore, the thermal conductivity in the liquid state must be quite large.

In this study, only two temperature profiles were measured and used to estimate the temperature-dependent thermal conductivity; one (at r=5 mm) was used in the inverse analysis, while the other (at r=21 mm) served as a Dirichlet boundary condition. In contrast, a Neumann (rather than Dirichlet) boundary condition is usually used in the literature [26,27]. However, the heat flux at r=21 mm is unknown, meaning that a Neumann boundary condition cannot be directly applied. An alternative is to simultaneously estimate both the temperature-dependent thermal conductivity and boundary (at r=21 mm) heat flux, and the two temperature profiles (at r=5 mm and r=21 mm) can then both be used in the inverse analysis. To implement this, a third-order polynomial is still used to approximate the thermal conductivity, while a second-order polynomial is used to approximate the temperature-dependent boundary heat flux:(19)$$q\left(T\right)=Q_{1}+Q_{2}T+Q_{3}T^{2},$$
where *q* is the boundary heat flux; $Q_{1}$, $Q_{2}$, and $Q_{3}$ are unknown parameters; *T* is the boundary temperature (at r=21 mm). The corresponding Dirichlet boundary condition (Equation (Equation 3)) in the direct problem is replaced with the Neumann boundary condition
(20)$$-k\left(T\right)\frac{\partial T\left(r,t\right)}{\partial r}=q\left(T\right)\;\mathrm{for}\;r=r_{0}.$$

Furthermore, the sensitivity coefficients for the unknown parameters $Q_{1}$, $Q_{2}$, and $Q_{3}$ can also be obtained by the boundary-value method (see Section 2.2), and the corresponding governing equations are given by
(21)$$\begin{array}{cccc} \rho \left(T\right)c\left(T\right)\frac{\partial J_{j}^{\prime }\left(r,t\right)}{\partial t} & =k\left(T\right)\frac{\partial^{2}J_{j}^{\prime }\left(r,t\right)}{\partial r^{2}}+\left[\frac{\partial k\left(T\right)}{\partial T}\frac{\partial T}{\partial r}+\frac{k\left(T\right)}{r}\right]\frac{\partial J_{j}^{\prime }\left(r,t\right)}{\partial r} & \; & \;\mathrm{for}\;0<r<r_{0},\;t>0,\\ \frac{\partial J_{j}^{\prime }\left(r,t\right)}{\partial r} & =0 & \; & \;\mathrm{for}\;r=0,\;t>0,\\ -k\left(T\right)\frac{\partial J_{j}^{\prime }\left(r,t\right)}{\partial r} & =\frac{\partial q\left(T\right)}{\partial Q_{j}} & \; & \;\mathrm{for}\;r=r_{0},\;t>0,\\ J_{j}^{\prime }\left(r,t\right) & =0 & \; & \;\mathrm{for}\;0\leq r\leq r_{0},\;t=0, \end{array}$$
where $J_{j}^{\prime }=\partial T/\partial Q_{j}$(j=1,2,3). If the vector of unknown parameters (see Equation (Equation 2)) $K={\left[K_{1}\;K_{2}\;K_{3}\;K_{4}\right]}^{T}$ is replaced by $P={\left[K_{1}\;K_{2}\;K_{3}\;K_{4}\;Q_{1}\;Q_{2}\;Q_{3}\right]}^{T}$, then the present inverse analysis method can be extended to simultaneously estimate both the thermal conductivity and the boundary heat flux. As shown in Figure 12, both the recovered temperature profiles (at r=5 and 21 mm) are in good agreement with the measured temperature profiles (Figure 12a). The boundary heat flux decreases rapidly with decreasing boundary temperature (Figure 12b). The temperature-dependent thermal conductivity estimated using the Neumann boundary condition is also in good agreement with that obtained using the Dirichlet boundary condition (Figure 12c).

## 4. Conclusions

In this study, a Gauss–Newton algorithm was used with an inverse analysis method to estimate the temperature-dependent thermal conductivity of a DNAN/HMX melt-cast explosive with the help of known temperature-dependent density, specific heat capacity, and other thermophysical properties. This method needs only two experimental temperature profiles; one is used in the inverse analysis, and the other serves as a boundary condition. The temperature-dependent thermal conductivity of this explosive can be approximated using a polynomial whose coefficients can be estimated by the present method.

This inverse analysis method was firstly verified by simulated measurement temperatures with two types of prespecified temperature-dependent thermal conductivity. Type one was a first-order (linear) polynomial, and type two was a second-order (quadratic) polynomial. Regardless of the type of polynomial used, and no matter how large the random noise (the maximum standard deviation was 0.5 ${}^{\circ }C$) in the simulated measurement temperatures, the estimated thermal conductivity was in good agreement with the prespecified values, demonstrating the effectiveness of the present inverse analysis method.

The estimated temperature-dependent thermal conductivity of the DNAN/HMX melt-cast explosive is approximately constant when the temperature is below the solidus line, and it increases rapidly with increasing temperature above the solidus line. The polynomial order (greater than or equal to three) has little effect on the estimated thermal conductivity when the temperature is above the lowest experimental temperature value used in the inverse analysis method and below the initial high temperature of this molten explosive. The aforementioned law is consistent whether only the thermal conductivity is estimated or both the thermal conductivity and the boundary heat flux are simultaneously estimated. It should be noted that, however, the present method did not investigate the effect of the flow of the molten DNAN/HMX explosive on the estimated thermal conductivity of this explosive. In fact, flow effects are important when estimating the thermal conductivity of Comp-B (a classical melt-cast explosive) [11].

However, whether or not flow effects are included, the thermal conductivity of Comp-B in the liquid state is higher than that in the solid state [11]. Similarly, the present estimated thermal conductivity of the DNAN/HMX explosive in the liquid state is higher than that in the solid state. This fact, together with other results and discussion above, demonstrates that the present inverse analysis method can provide a good approximation when estimating the thermal conductivity of melt-cast explosives.

In future work, to generalize the present method to the estimation of the temperature-dependent thermal conductivity of other melt-cast explosives, flow effects should be included in the inverse analysis method, and more experiments should be conducted—in particular, the thermal conductivity of high-temperature molten explosives should be measured—and the results should be compared.

## Figures and Tables

**Figure 1 materials-15-02077-f001:**
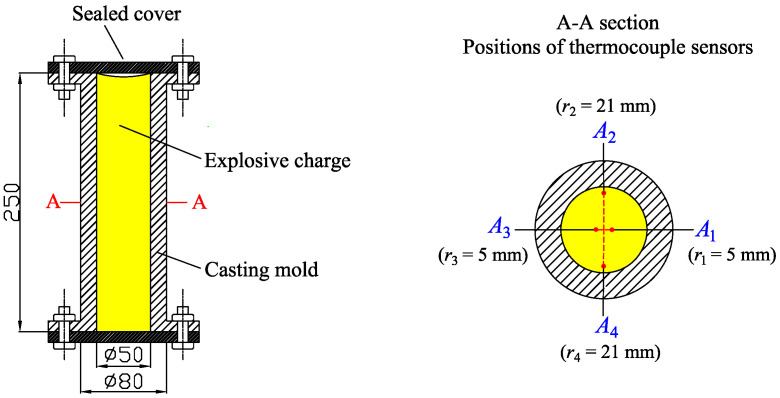
Schematic of experimental setup.

**Figure 2 materials-15-02077-f002:**
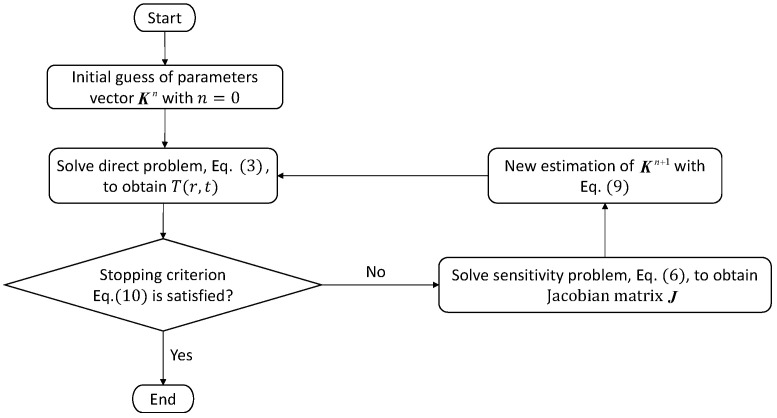
Flow chart of iterative procedure.

**Figure 3 materials-15-02077-f003:**
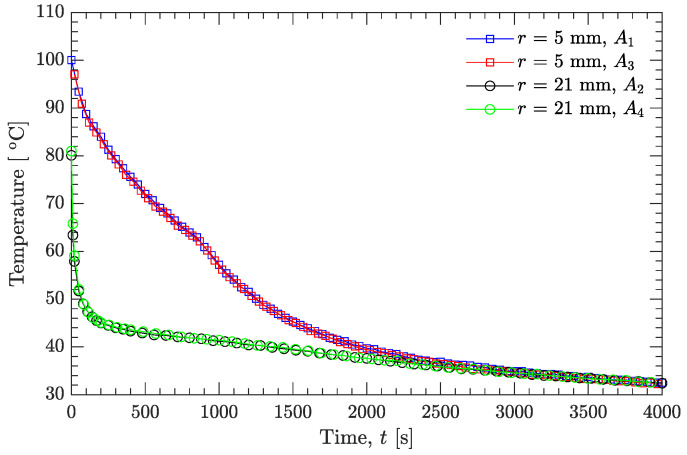
Experimental temperature profiles.

**Figure 4 materials-15-02077-f004:**
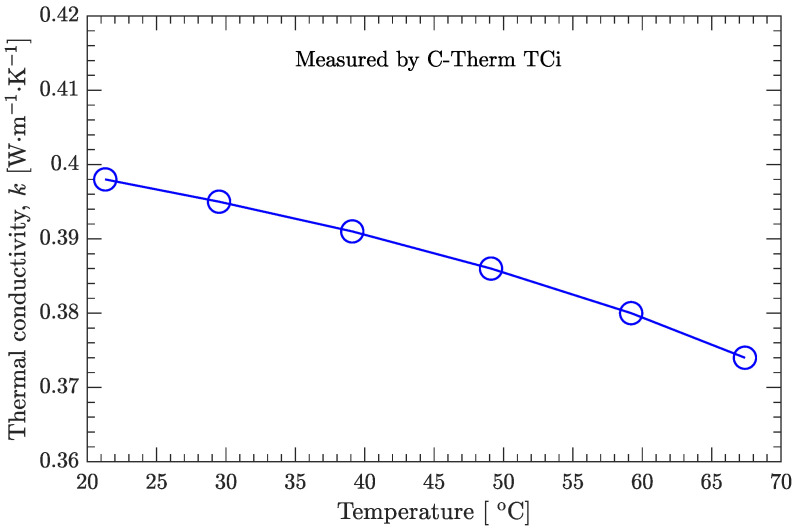
Thermal conductivity of the DNAN/HMX melt-cast explosive in the solid state.

**Figure 5 materials-15-02077-f005:**
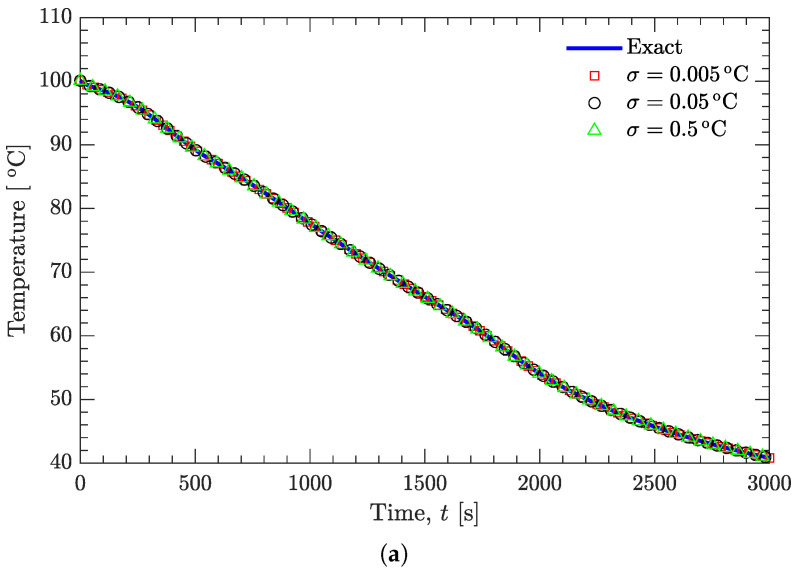
Effect of random noise on the recovered temperatures and on the estimated thermal conductivity; a linear function of k(T) is prespecified. (**a**) Temperature. (**b**) Thermal conductivity.

**Figure 6 materials-15-02077-f006:**
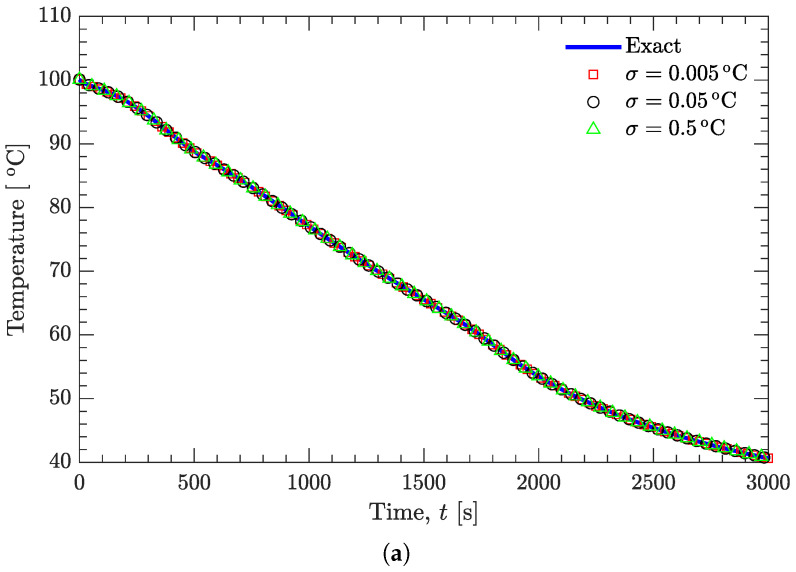
Effect of random noise on the recovered temperatures and on the estimated thermal conductivity; a quadratic function of k(T) is prespecified. (**a**) Temperature. (**b**) Thermal conductivity.

**Figure 7 materials-15-02077-f007:**
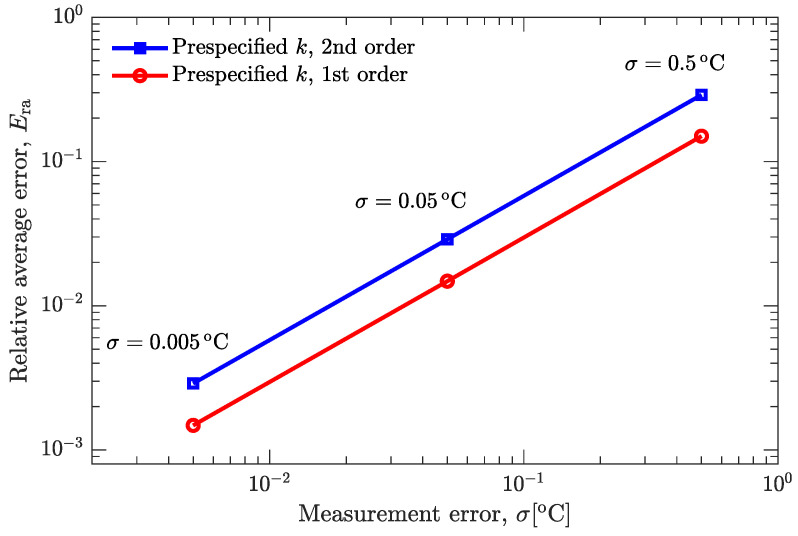
Effect of random noise on the relative average error.

**Figure 8 materials-15-02077-f008:**
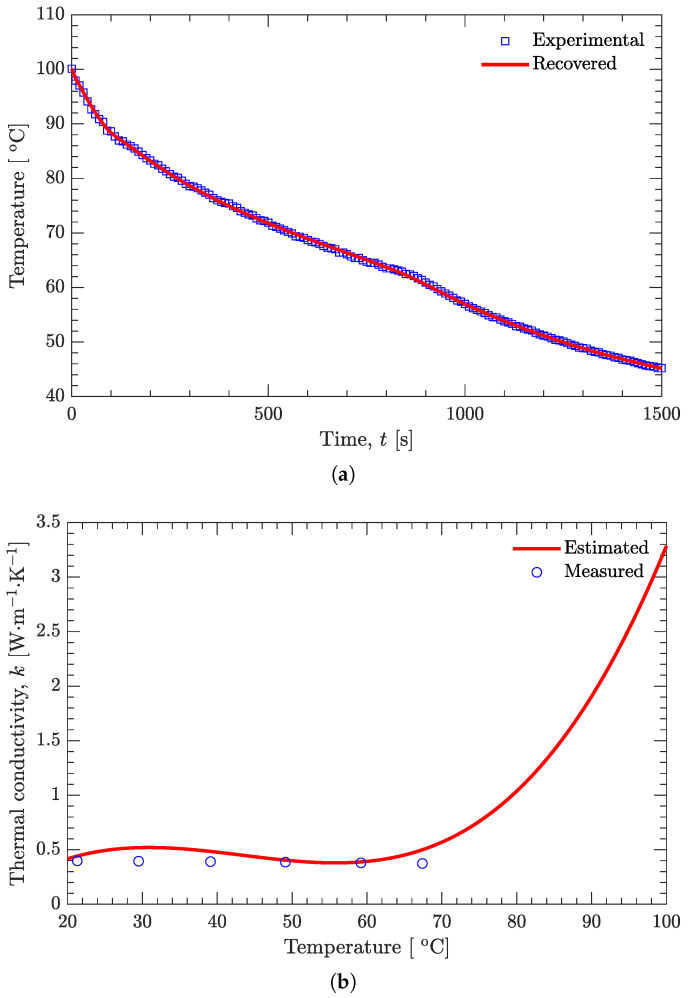
Recovered temperature and estimated thermal conductivity based on the experimentally measured temperatures of the DNAN/HMX melt-cast explosive. (**a**) Temperature. (**b**) Thermal conductivity.

**Figure 9 materials-15-02077-f009:**
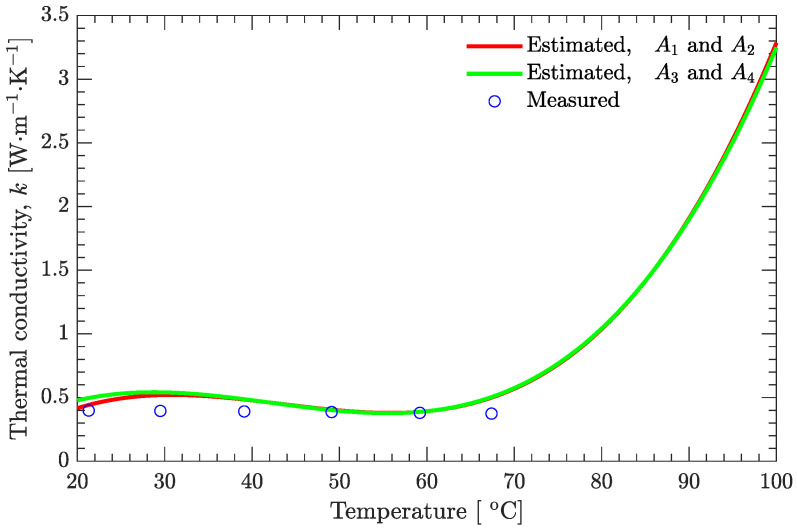
Effect of sensor location errors on the estimated thermal conductivity.

**Figure 10 materials-15-02077-f010:**
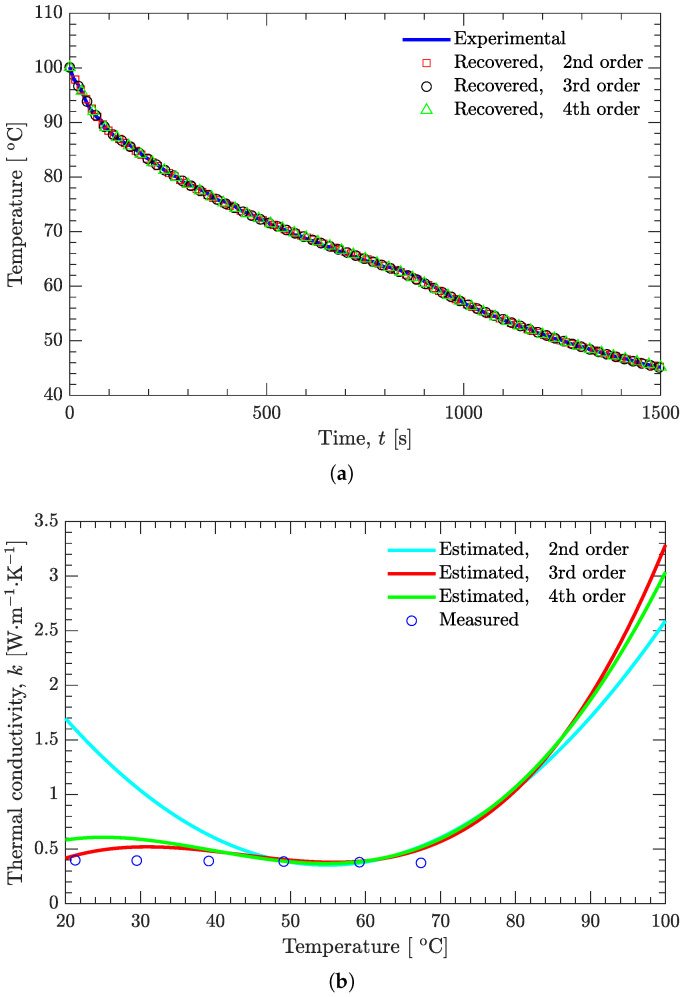
Effect of polynomial order on the recovered temperatures and on the estimated thermal conductivity. (**a**) Temperature. (**b**) Thermal conductivity.

**Figure 11 materials-15-02077-f011:**
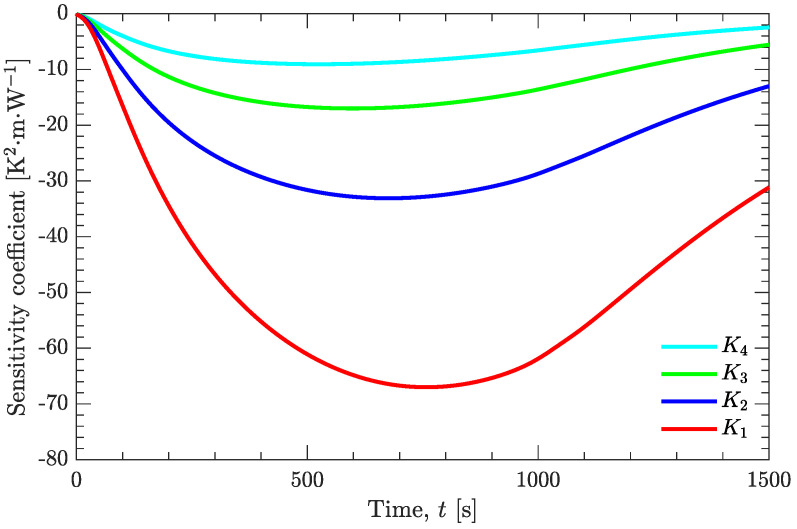
Sensitivity coefficient of the unknown parameters to be estimated.

**Figure 12 materials-15-02077-f012:**
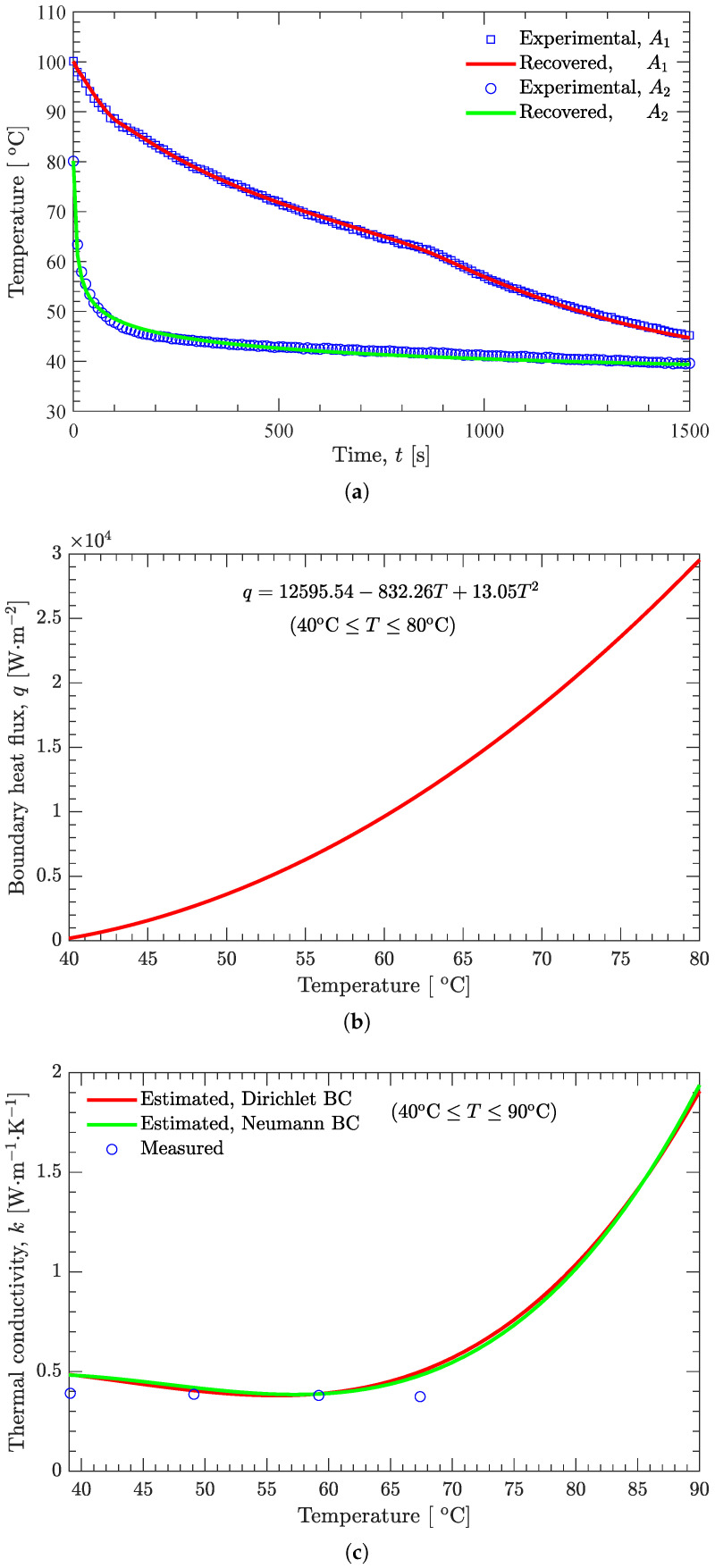
Effect of boundary-condition type on the inverse analysis. (**a**) Temperature. (**b**) Heat flux. (**c**) Thermal conductivity.

## Data Availability

The data presented in this study are available on request from the corresponding author.

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
