# Peer review of "An Inverse Analysis for Establishing the Temperature-Dependent Thermal Conductivity of a Melt-Cast Explosive across the Whole Solidification Process"

_materials, 2022, doi:10.3390/ma15062077_

Round 1

Reviewer 1 Report

The paper deals with  inverse analysis for establishing the temperature-dependent thermal conductivity of a melt-cast explosive. This is very relevant subject. The results of conducted research are very useful and have good practical application. The paper is very well written. The structure is logical and corresponds to IMRAD profile however literature review should be extended as there are more new studies in this field. The results of this study should be discussed and compared in the light of other new studies in this field. Conclusions need to be strengthen with practical implications. Also the limits of conducted research should be clearly identified and future research guidelines need to be provided. 

Author Response

Thanks to the referee for their recognition f the manuscript and valuable suggestions. Regarding the literature review, we added four references in relevant fields in the recent five years in the literature review section, among which two (8,9) are about the thermal conductivity test methods, (line27-29) and two (20,21) are about the application examples of IHTPs. (line44-45) For the results and discussion section of this study, compared with the content of reference 11, the results show that the thermal conductivity of liquid-cast explosives is higher than that of the solid state without considering the flow. which indicates that the results of this study are reliable and can provide reference for the thermal conductivity test of melt-cast explosives. (line 289-293) At the same time, we pointed out the limitations of this study and the future research direction. In the future work, the influence of flow on the thermal conductivity can be considered in the model, and the thermal conductivity of explosives at high temperature can be measured and further compared with the model calculation results. (line294-304) Finally, I would like to thank the referee for their works.

Reviewer 2 Report

The manuscript still has some errors in English grammar as well as many descriptions are difficult to understand, so it is recommended that the authors review and correct it.

Author Response

We are very sorry for some minor mistakes in the manuscript that should not have occurred due to our negligence. We read the manuscript carefully and made careful revisions to the language. We have revised two places in the article altogether. We substituted tetramethylenetetranitramine for cyclotet-ramethylenetetranitramine in the abstract and introduction. (line 5, 30) At the same time, we changed "However" to "Besides" in line 113 to make it easier for readers to understand. (line 113). Special thanks to you for your good comments.

Reviewer 3 Report

The manuscript is written well; the references are appropriate.

I was hoping to see the deviation in temperature from thermocouple measurements.  For example,

in "The molten DNAN/HMX explosive was prepared with an initial temperature of 94
100.1C." what is the accuracy in temperature?

It would help to address the temperature accuracy and deviation. 

I recommend the manuscript for publication in Materials.

Author Response

Thank you very much for your suggestions.

Regarding ‘which is the accuracy in temperature?’ in your suggestion. As shown in the manuscript below, the exact temperature is 100.1 °C, and the 94 is the line number. For more information, please refer to reply review report.

Finally, thank you again for your valuable advice and good luck with your work!

Reviewer 4 Report

Here is the evaluation:
1. In my point, the Figure 12 comes before the conclusions.
2. In pg2 and line 86 45# steel should be briefly described.

  1. Check the for the "°C" through the whole paper, space is needed between "for examples "21.1" and "°C".

Author Response

Thank you very much for your valuable suggestions.

I have made the following revisions to your three suggestions, and I appreciate your suggestions for making my manuscript better.Details can be found in reply review report!

  • As you said, Figure 12 should be placed before the conclusion, we have made changes, we deal with it like this in the latex document. But the latex template puts it in the conclusion, we are very sorry for that, I believe that figure 12 can be put before the conclusion in the layout of the article before publication.
  • Based on your suggestion, we have added a description of No. 45 steel to the article as follows. (0.42-0.50 wt% C, 0.50- 860.80 wt% Mn, ≤ 0.035 wt.% P, ≤ 0.035 wt% S)(line 86)
  • We have double checked and corrected your question, adding a space before each °C. Thank you for your careful and careful examination of the manuscript.

Finally, thank you for your suggestions on the manuscript again, which have benefited me a lot.  I wish you have a happy life and smooth work!
